# Creating simple predictive models in ecology, conservation and environmental policy based on Bayesian belief networks

**Victoria Dominguez Almela**[1], **Abigail R. Croker**[2¤], **Richard Stafford**[3]*

**1** School of Geography and Environmental Sciences, University of Southampton, Southampton, United Kingdom, **2** Centre for Environmental Policy, Imperial College London, London, United Kingdom, **3** Department of Life and Environmental Sciences, Bournemouth University, Poole, United Kingdom

¤ Current address: Centre for BioComplexity, High Meadows Environmental Institute, Princeton University, Princeton, NJ, United States of America

* rstafford@bournemouth.ac.uk

**Data Availability Statement:** All data is included as supplementary material. All R code is available on Github at https://github.com/vda1r22/bbnet.

## Abstract

Predictive models are often complex to produce and interpret, yet can offer valuable insights for management, conservation and policy-making. Here we introduce a new modelling tool (the R package '*BBNet*'), which is simple to use, and requires little mathematical or computer programming background. By using straightforward concepts to describe interactions between model components, predictive models can be effectively constructed using basic spreadsheet tools and loaded into the R package. These models can be analysed, visualised, and sensitivity tested to assess how information flows through the system's components and provide predictions for future outcomes of the systems. This paper provides a theoretical background to the models, which are modified Bayesian belief networks (BBNs), and an overview of how the package can be used. The models are not fully quantitative, but outcomes between different modelled scenarios can be considered ordinally (i.e. ranked from 'best' to 'worse'). Parameterisation of models can also be through data, literature, expert opinion, questionnaires and/or surveys of opinion, which are expressed as a simple 'weak' to 'very strong' or 1–4 integer value for interactions between model components. While we have focussed on the use of the models in environmental and ecological problems (including with links to management and social outcomes), their application does not need to be restricted to these disciplines, and use in financial systems, molecular biology, political sciences and many other disciplines are possible.

## 1. Introduction

The paucity of quantitative skills in the environmental workforce and among graduate students is well documented [1, 2]. Meanwhile, quantitative ecology continues to create and develop increasingly sophisticated models, embracing complex mathematics and AI principles [3, 4]. This creates difficulties for many environmental professionals; not only is modelling not an available tool without employment of specialists, but the complexity of the models and their

**Funding:** The author(s) received no specific funding for this work.

**Competing interests:** The authors have declared that no competing interests exist.

outcomes often makes it hard to convince decision makers and end users of their validity. This is especially true for AI approaches like Artificial Neural Networks, which lack transparency in how predictions are made [5–7].

Unlike some of the complexity of ecological models, environmental policy's evidence needs are often quite basic (e.g. ensuring a situation does not get worse, or a population is on an increasing trajectory). Such coarse levels of prediction can also be useful to ecologists and conservationists who may subsequently try to validate models through data collection and experimentation [6, 8, 9]. However, policy and conservation decisions are rarely made in isolation. The effects on other components of the wider 'system', including the rest of the ecosystem, ecosystem services, local communities, employment, and health, also need to be considered [10, 11].

Bayesian belief networks (BBNs) are tools which can be used to model system behaviours and have been used in a number of ecological applications [8, 12–16]. Technically they are probabilistic graphical methods, more simply, they model complex systems through probabilities assigned to different components of a system (e.g. species in a foodweb) and the interactions between these components (i.e. trophic interactions between species) [8]. They are capable of using a variety of information sources in their design and parameterisation, from field data through to qualitative data and expert opinion [8, 17]. As such, they can be useful tools to model understudied systems, or to study interactions between systems (such as interactions between ecological and social systems) [15, 17, 18].

However, the complexity of most systems means that multiple interactions and multiple drivers are frequently affecting any single component of a system (e.g. a given species might be competing with many others for resources, as well as feeding on a variety of species, and being predated by many others).With traditional BBNs, such complexity requires complex model parameterisation and building models can become overwhelming and impossible to populate beyond just (largely uneducated) guesswork. Furthermore, the inability of reciprocal feedback between network nodes (i.e. reciprocal competition between species, or the consideration of both bottom-up and top-down processes acting simultaneously) and inability to construct feedback loops also limit their use in ecological disciplines [19]. In complex systems, there is also a tendency for 'signal loss' as signals or changes propagate through the network, meaning that the predicted effect of a change becomes smaller and smaller until it is almost indetectable, making it difficult to interpret the outcomes of the models [12, 19].

More recent work has modified these BBNs approaches by simplifying the development of models with complex interactions and implementing programming loops to determine reciprocal interactions. Additionally, automated computer decision making has been used to ensure signals propagate through the network [8]. Computational methods to help estimate uncertainty have also been incorporated in some models [20]. The models have been used on a variety of ecological and socio-ecological systems and to help examine the effects of environmental policies at local and national/international levels [20, 21]. Furthermore, while software related issues still arose, the fundamental principles of these models and their construction and parameterisation did not require detailed modelling knowledge. These tasks could be successfully achieved within a few hours by first year undergraduate students [22]. As such, the 'BBNet' package puts tools for constructing and interpreting ecological models in the hands of a far broader number of environmental scientists and professionals than has previously been the case.

The purpose of this paper is to present (1) the underlying theory of the modified Bayesian belief networks, (2) introduce the 'BBNet' package as a user-friendly interface for ecological and environmental researchers and practitioners with limited modelling experience to produce useful and meaningful models, and (3) suggest a workflow for the formulation of these models, including parameterisation of the model and dealing with uncertainty.

## 2. Theoretical basis

Bayesian belief networks (BBNs) are a modelling approach where interactions between different components of complex systems can be examined and predictions can be made for components of interest in these systems, as such, they can be used to make environmental or ecological predictions. For example, in foodwebs with multiple interacting species, an increase in the population size of one species can impact the entire ecological community, and relative changes to each population can be predicted by the model. However, models are not limited to foodwebs. They can also be used to investigate the effects of biological, economic, or policy changes on species, ecosystem functions, ecosystem services, and socio-economic outcomes (examples of these are in the references above).

More technically, BBNs create models based on causal graphs. Essentially a series of nodes (which may represent aspects of interest in the model, e.g. species, ecosystem services, laws, social outcomes) are connected by directional edges (direct relationships between the aspects of interest or between individual nodes). The relationship between nodes is defined by the edges–a fixed parameter of how the child node will respond if the parent node changes. These relationships are based on Bayesian inference, although non-Bayesian processes are also used to allow processes such as feedback loops and reciprocal interactions and to prevent signal loss (see below in the current section). Only direct cause and effect relationships are defined by the edges, indirect effects are an outcome of the modelling process. The theoretical basis for the model is based on that in Stafford et al. (2015) [8] and is described below. A number of updates and additional useful tools are provided in the '*BBNet*' package, described in the functions below (section 3), which provide additional functionality to understand and visualise the models and to examine information flow through the models.

Within the '*BBNet*' package each edge in the is given network an integer value between -4 and 4 to indicate the belief that a specific child node may increase or decrease, given an increase in the parent node. Negative numbers for edges equate to a mathematical negative relationship between nodes–i.e. an increase in the parent node will lead to a decrease in the child node. Positive numbers for edges equate to a mathematical positive relationship between nodes—i.e. an increase in the parent node will lead to an increase in the child node. A value of 0 does not need to be used for edges, as essentially the edge can be removed from the network.

Nodes are also given values between -4 and 4. These are the 'prior' values of each node, and these values can change as the model runs (unlike edge values, which do not change). Negative values equate to a reduction in the node (e.g. if the node represents a species, a negative value would indicate a decline in the population of the species). Positive values represent an increase in the node (e.g. an increase in population size). In complex social-ecological systems, there tends to be greater certainty over large events and their impacts, and greater uncertainty over smaller events and their emergent properties. Therefore, a value of 4 indicates high certainty over a greater magnitude of change in each node, and a value of -4 indicates low certainty over a lesser magnitude of change (see Table 1 for details of determining parameters for 'prior' nodes and edges). Prior values are only set for nodes where known changes will occur–e.g. if an intervention to cull a species was proposed, only the species culled would have a 'prior' value. Other nodes would be left with no prior knowledge (values of 0) and the effects on these nodes would be calculated by the model.

In determining edge values and prior node values, thought should be given to the spatial and temporal aspects which require modelling. The model has no direct temporal or spatial components (although an order of events can be investigated using some of the functions below). Temporal and spatial dimensions need to be considered in the edge and prior values, with an awareness that these may need to be changed if the temporal or spatial constraints of

**Table 1. Parameterisation values of edges and priors in the model.**

| Input value | Edge values | Prior values |
|---|---|---|
| 4 (or -4) | Strong relationship between parent and child node, creating a clear and noticeable cause and effect relationship. Full (> 95%) agreement between sources for the relationship | Full or large magnitude implementation of a change (i.e. doubling a large population size, increasing costs by 70–100%). It would be difficult to implement the change in greater detail |
| 3 (or -3) | Strong relationship between parent and child node, creating a clear and noticeable cause and effect relationship. Good agreement between sources for the relationship (>75% of data agree) OR Moderate relationship between parent and child nodes. Difference is detectable but may not be obvious. Full agreement between sources for the relationship | Moderate to large scale implementation of a change–i.e. removing 50% of a moderately abundant population |
| 2 (or -2) | Moderate relationship between parent and child nodes. Difference is detectable but may not be obvious. Good agreement between sources for the relationship (>75% of data agree) OR Weak relationship between parent and child nodes. Difference is apparent in studies but might not always be significant (i.e. due to low sample size). Full (> 95%) agreement between sources for the relationship | Small to moderate change. e.g. deer culling to remove 10% of deer |
| 1 (or -1) | Weak relationship between parent and child nodes. Difference is apparent in studies but might not always be significant (i.e. due to low sample size). Good agreement between sources for the relationship (>75% of data agree) | Smaller than above |
| 0 | No relationship, or large disagreement between sources | No direct change |

the model change. A biological example of temporal and spatial consideration is given in the case of starvation in the description of the rocky shore model below (section 3.1.1). In this example, small changes in species numbers will be important due to the limited spatial component of the model (communities are on isolated boulders), yet the limited duration of the model means that while grazing may have top down effects, starvation (a bottom up effect) is unlikely to have an effect on predators and grazers, and these interactions (or potential edges) are not included in the model. Other examples considering spatial and temporal aspects could include comparison of wildfires vs. controlled burning. Over a short timescale (i.e. days), and a small spatial area (e.g. the area of a controlled burn), both will have similar effects on the ecological communities, decimating biodiversity which was present. However, at a larger spatial scale, controlled burning may have much less impact than an uncontrolled fire. At longer spatial scales (months to years) the effects on biodiversity will also change (for example, there may be benefits of fire to biodiversity).

The use of integer values between -4 and 4 are added for purposes of clarity in building the model and are transferred to a value between 0 and 1 for the purposes of calculations. $P(Xi)$ (the probability of the node increasing) is derived from the integer values from -4 to 4 (Table 2). Note, that due to there always being some uncertainty in complex systems, both in terms of knowing a node will increase or decrease, and in terms of interactions between nodes, probabilities of both priors and edges have maximum values of 0.9 and minimum values of 0.1, rather than 1 and 0.

**Table 2. Transformations of prior node values and edge strengths from inputted values to those used for calculations.**

| Input value | Value used in calculations for increase |
|---:|---:|
| -4 | 0.1 |
| -3 | 0.2 |
| -2 | 0.3 |
| -1 | 0.4 |
| 0 | 0.5 |
| 1 | 0.6 |
| 2 | 0.7 |
| 3 | 0.8 |
| 4 | 0.9 |

In the following equations, the probability of a node decreasing ($P(X_d)$) is calculated by Eq 1:

$$P(X_i) + P(X_d) = 1 \tag{1}$$

With subscripts i and d indicate increasing or decreasing respectively for the nodes.

Intermediate probabilities of each node *increasing* given the different interactions from all connecting nodes are calculated using the following Bayesian equation:

$$P(X_i|Y) = [P(X_i|Y_i)*P(Y_i) + P(X_i|Y_d)*P(Y_d)] \tag{2}$$

where X is the node under consideration (the child node), and Y are the interacting nodes (parent nodes, considered one at a time). These values are calculated for each interacting node.

Where there is no knowledge of a change in value of node Y (i.e. the prior probability of change is 0.5) then this node is not included in the above equation (however, such inclusion might occur in future iterations of the model where the value of the node may have changed).

At this point, no 'prior' information on node X is included in the calculation. To ensure any prior knowledge available is maintained in the network, and to allow reciprocal interactions and feedback loops, the overall posterior probability for each node is calculated in two ways, the first ensuring that additional information on node interactions add to the certainty provided by the prior, the second will ignore prior values if information on species interactions provide more certain information (i.e. a value further away from 0.5) than the prior:

$$Post(X_i) = P(X_i) + |1 - P(X_i)|*[\sum_{1-n}(P(X_i)*(P(X_i|Y) - 0.5))/n] \tag{3}$$

and

$$Post(X_i) = [\sum_{1-n}(P(X_i|Y))]/n \tag{4}$$

where n is the number of interactions with species X. The final value of $Post(X_i)$ is given by the value displaying the most certainty (i.e. furthest in magnitude from 0.5). The model is then repeated for further iterations to allow information to propagate through the network, but with updated prior probabilities such that:

$$P(X_i) = Post(X_i) \tag{5}$$

The model then runs through additional iterations. When all iterations of the model are completed (four iterations are included in the bbn.predict() function, some functions allow this to be altered), conversion back to a -4 to 4 scale occurs using the following equation (note,

these final posterior values are not integers):

$$\text{Final change} = 10 \times (\text{Post}(X_i) - 0.5) \qquad [6]$$

Importantly, only nodes with *known* prior changes are altered in any scenarios provided at the start of a model (see also section 4.4). For example, if simulating a manipulative ecological experiment where a species of grazer was removed from an area, only the prior for this species of grazer would be altered, with the model calculating the predicted changes to other species based on the edge values already assigned to interacting nodes.

Technically, BBNs calculate the probability of a node increasing or decreasing. However, given difficulties in distinguishing probabilities (i.e., belief or certainty of a node increasing or decreasing) from magnitude for most natural phenomena (see [20]), the inclusion of Eqs 1 and 2 above disrupt the pure calculation of probability and help prevent signal loss through the network, allowing for more meaningful predictions. The conversion back to -4 to 4 reinforces this amalgamation of probability and magnitude, by not presenting the data as a probability. While this conversion means that model outputs cannot be treated as interval or ratio data (i.e., you cannot numerically measure the differences between values -4 to 4), these values can be compared across different models and act as ordinal variables as a minimum (i.e. different scenarios can be ranked by changes to variables of interest, as per [21]).

## 3. The '*BBNet*' package

The '*BBNet*' Package consists of a series of functions to create and obtain results from causal graph models, as well as two examples of systems with various implemented scenarios. The example datasets are discussed, followed by each of the package functions. Additional information, beyond the basis of the model described in section 2 above, is provided below, where it relates to particular functions in the package.

The '*BBNet*' package is available from both CRAN and GitHub (https://github.com/vda1r22/bbnet).

You can install the stable version of '*BBNet*' from CRAN or GitHub with:

*install.packages("bbnet")* or

devtools::install_github("vda1r22/bbnet")

All data files discussed are available as datasets within the package, or as.csv files in S1 File. Further information on datasets and requirements are provided in a video tutorial (S2 File). A video tutorial on running the model is provided in S3 File. The R markdown script used in the video is provided in S4 File.

### 3.1 Example datasets

**3.1.1 Rocky shore ecology.** The BBN model here uses the interactions described previously [8]. It is a simple model of rocky shore interactions (trophic interactions and competition) between species on isolated boulders on a rocky shore. It is designed for scenarios relating to experimental manipulation of predator and grazer abundance on isolated boulders over a 4–8 week period, and as such, trophic interactions are top down only (starvation is unlikely to occur in this time period, but populations are small, given the spatial isolation of the community on boulders–see discussion in [8]). Five csv files are provided (S1 File). (1) RockyShoreNetwork.csv provides the edge strengths for the network. These are given as values between -4 to +4 (converted as per Table 1 before running the model) and represent the probability of a child node *increasing* given that the parent node was *increasing* (with a value of 1 after conversion to 0 to 1 values). In this file, when opened in a spreadsheet, the node listed at

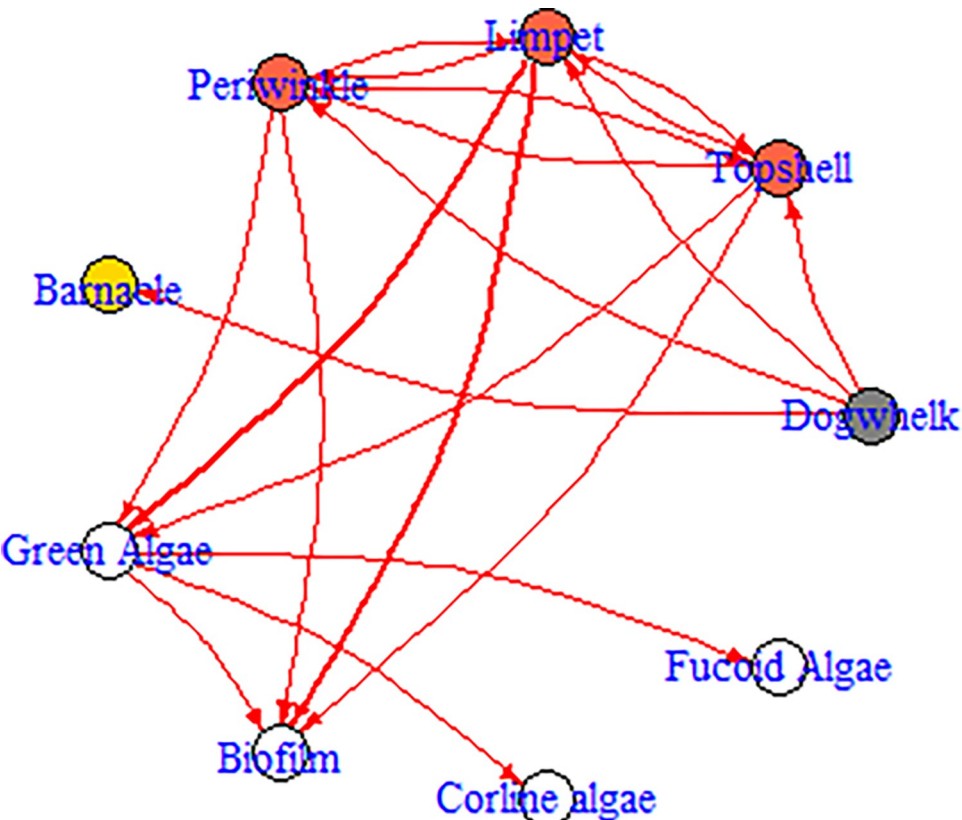

**Fig 1. Interaction diagram of the rocky shore model, produced by the `BBN.visualise()` function.** Nodes are colour coded to represent functional groups (white = algae, grey = predator, orange = grazers, yellow = filter feeders). Arrows point from the parent node to the child node. Red arrows indicate negative interactions between nodes. Black arrows (not present in this figure) represent positive interactions.

the start of each *row* affects the indicated species in each *column* (see S2 File, creating input files). (2) Dogwhelk_Removal.csv is a scenario for the model, and provides initial prior values for each node (note, all nodes have the value of 0, or no change, other than dogwhelks)–this scenario represents a removal of all dogwhelks from the area. (3) Winkle_addition.csv is a scenario node representing the addition of periwinkles to an area. (4) Combined_treatment.csv represents a removal of dogwhelks and addition of periwinkles to an area. (5) RockyShoreNetworkDiagram.csv is a slightly altered version of the edge strengths model -see above–in this case, it contains additional parameters for use with the `BBN.network.diagram()` function and has been used with this function to produce Fig 1.

**3.1.2 MPA management.** This example contains 3 data files (S1 File). The network model here (MPANetwork.csv) is based around a simple foodweb in a Marine Protected Area (MPA), but also includes human activities (fishing and scuba diving) and an overall indication of revenue from the area (from fishing and diving activities). No management measures are included in the model, but the scenarios indicate how these can be implemented–i.e. a potting ban (NoPotting.csv) will reduce the lobster fishery node. A no take scenario (NoTake.csv) will affect both fishing nodes. Again, it is only the direct effects which are accounted for in the scenario nodes, the model determining the changes to other nodes (e.g. an increase in diving due to more fish and lobsters is *not* included as a model prior value in any scenario).

## 3.2 '*BBNet*' package functions

**3.2.1 `bbn.predict()` *making predictions, bootstrapping and outputting data*.** This is the main predictive function, requiring an interaction network in the format of a n+1 by n matrix or dataframe (where n is the number of nodes, row names form the first column, but the column names are imported as a header, resulting in the extra column of row names), indicating edge strengths between each node (see examples in sections 3.1 and S2 File for detailed requirements). It also requires between 1 and 12 scenarios (each scenario represented by a 2 x n dataframe where n is the number of nodes in the network) which show initial changes to some of the prior values of the nodes. All of these files should have integer values ranging between -4 and 4, indicating the degree to which the node increases (negative numbers therefore represent a decrease), with scenario files having values of 0 for nodes with no prior information, and the interaction network matrix having blank values where no edges connect nodes.

The function offers potential to 'bootstrap' the outputs of the model to understand inherent uncertainty. Such uncertainty can arise due to the exact parameterisation of edges and priors, and the fact that some interactions have larger changes in magnitude on child nodes than others. As such, bootstrapping allows the uncertainty of the predictions to be visualised as error bars. Bootstrapping involves randomly selecting and modifying edge strengths to determine overall changes to the posterior node values. This bootstrapping process is run multiple times (number determined by user–see *"boot_max"* below) and 95% confidence intervals of the output of each parameter are calculated by removing the highest and lowest 2.5% of values for each posterior node (as per methods in [23]). These confidence intervals are applied to the actual values calculated using unadjusted parameters. If bootstrapping is applied to the modelling process, then the first run through does not adjust any parameters and is displayed as the 'point' or filled circle in any figures produced, or the first column of any numeric output produced.

R Function and arguments

bbn.predict(BBN.model, priors1, . . ., boot_max, values, figure, font.size)

Required arguments

*bbn.model* - a matrix or dataframe of interactions between different model *nodes* (as described above)

*priors1* - an X by 2 array of initial changes to the system under investigation for a given scenario.

Optional Arguments

. . . *prior*S2 - *priors12* - as above, but additional scenarios.

*boot_max* - the number of bootstraps to perform. Suggested range for exploratory analysis 1–1000. For final analysis recommended size = 1000–10000—note, this can take a long time to run. Default value is 1, running with no bootstrapping—suitable for exploration of data and error checking.

*values* - default value 1. This provides a numeric output of posterior values and any confidence intervals. Set to 0 to hide this output.

*figure* - default value 1. Sets the figure options. 0 = no figures produced. 1 = figure is saved in working directory as a PDF file (note, this is overwritten if the name is not changed, and no figure is produced if the existing PDF is open when the new one is generated). 2 = figure is produced in a graphics window. All figures are combined on a single plot where scenario 2 is below scenario 1 (i.e. scenarios work in columns then rows).

*font.size* - default = 5. This sets the font size on the figures.

## Example

```
my_BBN <- read.csv('RockyShoreNetwork.csv', header=T)
  dogwhelk <- read.csv('Dogwhelk_Removal.csv', header = T)
  winkle <- read.csv('Winkle_addition.csv', header = T)
  combined <- read.csv('Combined_Treatment.csv', header = T)
  bbn.predict(bbn.model = my_BBN, priors1 = dogwhelk,
priors2 = winkle, priors3= combined, figure = 2, boot_max = 100,
values = 0, font.size = 7)
```

**3.2.2. `bbn.timeseries()` *understanding node behaviour over different timesteps.*** This function helps visualise the flow of information through the network and how changes progress through the network over time (e.g. changes occurring in one parameter before another—as per trophic cascade or ecological succession type processes). It should be noted that the exact values from these functions do not correspond to the more robust `bbn.predict()` which should be used to inform of likely changes (this function does not implement Eqs 2 and 3 detailed in the theoretical basis above).

As for `bbn.predict()` we need to pass the function a network model and a scenario as a minimum. In this case, only one scenario can be analysed at once. The output is a graph of each node in the network, visualised across the different timesteps in the model. Note—values are plotted on each graph and lines of best fit are drawn using the geom_smooth function. Typically this function may not perform well with the variability in values and lack of data points, and multiple warning messages may be produced, but the shape of the response is still visible.

R Function and arguments

`bbn.timeseries(BBN.model, priors1, timesteps, disturbance)`

Required arguments

`bbn.model` - a matrix or dataframe of interactions between different model *nodes* as per above.

`priors1` - an X by 2 array of initial changes to the system under investigation. The first column should be a -4 to 4 (including 0) integer value for each node in the network with negative values indicating a decrease and positive values representing an increase. 0 represents no change.

Optional Arguments

`timesteps` - default = 5. This is the number of timesteps the model performs. Note, timesteps are arbitrary and non-linear. However, something occurring in timestep 2, should occur before timestep 3.

`disturbance` - default = 1. 1—creates a prolonged or press disturbance as per the `bbn.predict()` function. Essentially prior values for each manipulated node are at least maintained (if not increased through reinforcement in the model) over all timesteps. 2—shows a brief pulse disturbance, which can be useful to visualise changes as peaks and troughs in increase and decrease of nodes can propagate through the network

*Example*

```
my_BBN <- read.csv('RockyShoreNetwork.csv', header=T)
  dogwhelk <- read.csv('Dogwhelk_Removal.csv', header = T)
  bbn.timeseries(bbn.model = my_BBN, priors1 = dogwhelk,
timesteps = 5, disturbance = 2)
```

**3.2.3. `BBN.visualise()` *visualising information flow through the network over model timesteps*.** This produces similar data to `bbn.timeseries()` (section 3.2.2) but in a very different visual format. A network diagram (similar to Fig 1) is produced, consisting of all

nodes. Each node is ordinally weighted with the biggest increase in black and the smallest increase (which potentially is a decrease) in white. Not all edges are plotted, only those exceeding a certain threshold. This allows the flow of information through the network to be visualised at each timestep more clearly.

R Function and arguments

```
bbn.visualise(BBN.model, priors1, timesteps, disturbance,
threshold, font.size, arrow.size)
```

Required arguments

`bbn.model` – a matrix or dataframe of interactions between different model *nodes*

`priors1`- an X by 2 array of initial changes to the system under investigation. The first column should be a -4 to 4 (including 0) integer value for each node in the network with negative values indicating a decrease and positive values representing an increase. 0 represents no change.

Optional Arguments

`timesteps` - default = 5. This is the number of timesteps the model performs. Note, timesteps are arbitrary and non-linear. However, something occurring in timestep 2, should occur before timestep 3.

`disturbance` - default = 1. 1—creates a prolonged or press disturbance as per the bbn. predict() function. Essentially prior values for each manipulated node are at least maintained (if not increased through reinforcement in the model) over all timesteps. 2—shows a brief pulse disturbance, which can be useful to visualise changes as peaks and troughs in increase and decrease of nodes can propagate through the network

`threshold` - default = 0.2. Nodes which deviate from 0 by more than this threshold value will display interactions with other nodes. As mentioned, values in these visualisation functions don't directly correspond to those in the *bbn.predict()*function. This value can be tweaked from 0 to 4 to create the most useful visualisations.

`font.size` - default = 0.7. Changes the font in the figure produced. The value here is a multiplier of the default font size used in the '*igraph'* package and does not correspond to the `font.size` argument in the `bbn.timeseries()` function.

`arrow.size` - default = 4. Changes the size of the arrows. Note, sizes do vary based on interaction strength, so this is a multiplier for visualisation purposes.

Example

```
my_BBN <- read.csv('RockyShoreNetwork.csv', header=T)
dogwhelk <- read.csv('Dogwhelk_Removal.csv', header = T)
bbn.visualise(bbn.model = my_BBN, priors1 = dogwhelk,
timesteps = 5, disturbance = 2, threshold=0.05, font.size=0.7,
arrow.size=4)
```

**3.2.4 `BBN.sensitivity()` running sensitivity analysis.** For some methods of model parameterisation, extensive data extraction from literature, or expert opinion can be useful. However, this is time consuming, and being aware of the most sensitive edge parameters in the model which may affect the desired outputs could help concentrate efforts. This function produces a list of the most important edge parameters (interaction strengths) that might require further examination, with importance increasing with numerical value (frequency number).

The function works by bootstrapping, consisting of multiple changes to prior values and interaction strengths in the network (the same process used for bootstrapping in the `bbn.predict()` function: selecting 10% of interactions in each iteration and adjusting them by a randomly determined amount of up to ± 0.1, based on the probability values, rather than the integer input values). The frequency value produced shows the number of times a modified

interaction shows up as important in causing a change to the listed nodes (the edge is counted as important each time it is changed and subsequently is in the 25% of bootstrapped cases which caused the biggest changes in the defined nodes of importance). As such, those interactions showing as more frequent in the table or figure are likely to be most influential in any predictions made. These edge values should be subject to closer scrutiny in terms of values used. Note, this does not mean the values are incorrect or should be reduced from more extreme values—i.e. from 4 to 3, just that they should be carefully checked, e.g. through literature searches, agreement amongst experts etc.

Required arguments

bbn.model - a matrix or dataframe of interactions between different model nodes

One or more nodes (recommended no more than 3) which would be the main outcomes of interest in the model. The spelling of these nodes needs to be identical (including capital letters) to that in the imported csv file (note, you should include spaces if these are in your csv file, rather than the dot notation used once imported into R)–see example below for more details.

Optional arguments

`boot_max` - the number of bootstraps to perform. Suggested range for exploratory analysis 100–1000. For final analysis recommended size = 1000–10000—note, this can take a long time to run. Default value is 1000.

Example

```
bbn.sensitivity(bbn.model = my_BBN, boot_max = 100, 'Limpet', 'Green Algae')
```

**3.2.5 `BBN.network.diagram()`** *creating a diagram of the network*.   This function visualises all nodes and interactions in a network, in a similar manner to the `bbn.visualise()` function (section 3.2.3), other than the full network, including all edges are shown. The strengths and directions of the edges are shown, but information 'flow' is not shown, and no scenarios are included in the function. Nodes can also be colour coded by theme. For simple models, this function can produce a visual representation of the model of interest, but for complex models, the visual representation is hard to interpret.

This function requires a slightly different input file, based on the normal BBN interaction model file. The first column is called id and consists of an '*s*' and a 2-digit number relating to the node number (e.g. s01, s02 and so on). The second column is called node.type and is an integer value from 1–4. This sets the colour of the node in the network (sticking to a maximum of four colours). For example, predators, grazers, filter feeders and algae could be colour coded separately. The third column is the same as the first column in the standard BBN interaction csv, other than it is titled node.name. It is important to use these column names (including capitals and dot notation). The remainder of the columns are exactly as the standard BBN interaction csv file (see S1 File in the Rocky Shore model for an example csv file or S2 File for further details of file requirements).

Required arguments

`bbn.network` - a csv file as described above, with note paid to the first three column names.

Optional arguments

`font.size` - default = 0.7. Changes the font in the figure produced. The value here is a multiplier of the default font size used in the '*igraph*' package and does not correspond to the `font.size` argument in the `bbn.timeseries()` function.

`arrow.size` - default = 4. Changes the size of the arrows. Note, sizes do vary based on interaction strength, so this is a multiplier for visualisation purposes. Negative interactions are shown by red arrows, and positive interactions by black arrows.

`arrange` - this describes how the final diagram looks. Default is `layout_on_sphere` but `layout_on_grid` provides the same layout as in the `bbn.visualise()` function and ensures nodes are structured in the order specified in the network. Other layouts, including `layout_on_sphere` are more randomly determined, and better/clearer diagrams may occur if you run these multiple times. Other options are from the '*igraph*' package:

```
layout.sphere
layout.circle
layout.random
layout.fruchterman.reingold
```

Examples

```
bbn.network.diagram(bbn.network = my_network, font.size = 0.
7, arrow.size = 4, arrange = layout_on_sphere)
bbn.network.diagram(bbn.network = my_network, font.size = 0.
7, arrow.size = 2, arrange = layout_on_grid)
```

## 4. Creating and parameterising '*BBNet*' models

There are many ways to create BBNs models, with differing degrees of time commitment and robustness, depending on the purpose of the final model. For example, models can be created and parameterised based on interactions assumed to be correct by the model developer. If these models were used to develop hypotheses to test experimentally, then this method would be suitable–empirical data would support or reject the models developed. A simple model, such as the rocky shore model discussed above, could be likely developed from 'best guess' estimates of the parameters in less than an hour. However, models used to make predictions which are not empirically tested may take much longer to develop and involve careful consideration over the nodes, edges, and interaction strengths. We discuss how to develop the model step by step, and considerations of each stage below.

### 4.1 Determining nodes

In some cases, such as for a species interaction web, determining nodes can be straightforward, as each node represents a species, or higher taxonomic group, in the area of interest. For example, in the rocky shore model provided [8] (Fig 1), the snail species were those commonly found on the boulders (other snail species were rare at <1% of total abundance). Seaweeds and barnacles were categorised on higher taxonomic classifications, with the assumption that all species within each grouping would respond in a similar manner to grazing pressure or competition. When wider environmental aspects, ecosystem functions and services, and socio-economics are added to models, the choice of nodes becomes more complex. Firstly, there will be output nodes—equivalent of dependent variables, or aspects of the system which need measurement. For example, this could be the relative abundance of a protected species, the economic value of an ecosystem service, or the amount of carbon sequestered within a habitat. Output nodes representing socioeconomic or cultural aspects that are not typically quantified require greater consideration, thinking about what an increase or decrease in this node represents in a meaningfully way. For example, a concept such as 'community acceptance' might be hard to quantify with traditional metrics, but the model will show if this is increasing or decreasing. There will also be clear input nodes which may have their prior values altered in the development of scenarios, such when exploring changes in policy and management, (e.g. preventing fishing in a marine protected area) or experimental manipulations (e.g. excluding grazing deer from a section of heathland). The intermediate nodes become a little more difficult to determine and relate to typical modelling issues of the need for sufficient detail. BBNs model direct interactions between nodes, so a direct

causal link should be established between all nodes in a model. However, as long as there is sufficient scope to include conflicts between different pathways, then the nodes can be quite broad scale. For example, an increase in mature tree coverage in an area is likely to lead to increased carbon sequestration, there would be no need to model photosynthetic pathways, for example. However, if the aim of a model was to address whether rewilding an area through natural succession was to increase carbon sequestration, then a direct link from rewilding to carbon sequestration would be incorrect. Rewilding may lead to various processes (including changes in predation and grazing) which may influence the amount of woodland, grassland, heathland, and other habitats in an area. A direct link between amount of woodland and carbon sequestration (and perhaps between grassland and heathland and carbon sequestration, but at different interaction strengths) can be made, but the amount of woodland will vary depending on other ecological factors.

Nodes must therefore capture the appropriate amount of detail needed for the model to be useful, without including excessive detail. For instance, if the link between amount of habitat and an ecosystem service are well established, but the mechanisms by which the habitat provides the ecosystem services are unclear, including the mechanism would reduce certainty and predictive power in the model and should, therefore, be avoided.

## 4.2 Determining edges

All nodes in the network should interact with other nodes via one or more edges. Unless a node is a clear 'output node' (see section 4.1), it should connect downstream to a child node. Equally, unless a node is a clear 'input node', it should act as a child node in the network. These rules, however, are not exclusive–an input node may be affected by another node in a network, and an output node can still be measured and go on to affect further nodes. Nodes can also have multiple edges as inputs or outputs. Edges are also directional. This means that node A can have an effect on node B, but node B will not have an effect on node A. Reciprocal interactions are possible (e.g. interspecific competition between species, where species A and B are nodes in the network) but are not required. For example, in the Rocky Shore model described (Fig 1), competition interactions are reciprocal, but trophic interactions were one way, with predators affecting prey only, due to the time over which the results were modelled (see [8] for details). In '*BBNet*' each edge acts independently on a node as per Eqs 4 and 5 (see section 2). This allows for much more complex networks to be built than traditional Bayesian belief networks, which require conditional probability matrices to be built when multiple edges act on a node. While some degree of control is lost in the model as a result, careful thought about model structure can overcome this (see S5 File). Finally, edges can represent either positive or negative interactions. These are defined mathematically, where a positive interaction creates a directional change in a child node in the same direction as the parent node (i.e. an increase in the parent node leads to an increase in the child node). A negative interaction creates a difference in direction between child and parent nodes (i.e. an increase in the parent node leads to a decrease in the child node). Care is needed here, especially when human-centric value judgements can be placed on the nodes. For example, increased use of fossil fuels has a [mathematically] positive effect on climate change (as one increases, so does the other). The models need this specified as a positive interaction although we tend to associate this as a negative outcome for society and the environment.

## 4.3 Determining edge strengths

Edges are given integer values between -4 and 4, where negative values indicate mathematically negative interactions between parent and child nodes. Values of zero indicate no interaction,

but for simplicity these should be left blank in the interaction matrix file. Decimal values will cause the '*BBNet*' package to crash and should not be used. The purpose of limiting interaction strengths to these integer values is to make the network easier to parameterise when limited information may be available. Given the output of the model can be described as 'ordinal' between scenarios, these levels of interaction strength are enough to provide clear differences between outputs and evaluate different scenarios. The `bbn.sensitivity()` function can also highlight parameters which cause the biggest differences to the outputs of the models, and therefore need the most data or highest certainty to parameterise (section 3.3.4; see S3 File for an example based on the rocky shore model).

It is possible to use published and grey literature to aid in the parameterisation process. Equally, quantitative or qualitative evidence from field or laboratory studies, interviews, focus groups, expert opinion, Delphi surveys can also inform the design and parametrisation of the models (section 4.5). The amount of evidence and agreement between studies, people or sources will help form the final values used in the model (for example, see [24] for a framework for a four stage degree of confidence framework). However, the magnitude of the change is also important. We suggest Table 1 is used to help formulate the edge and prior strengths.

## 4.4 Creating scenarios

Scenarios are changes to some of the nodes of a network. *Prior* node values are changed to integer values between -4 and 4 if these are directly manipulated or directly influenced nodes in a system. For example, in the rocky shore model, one scenario is the removal of dogwhelks from the system. All dogwhelks were removed, so the prior value was set to -4 (see Table 2). No further changes are made to the priors. The effect on dogwhelk removal on other aspects of the community are determined by the model as the numeric change in dogwhelks flows through the network. Another scenario involved removing dogwhelks and increasing periwinkles. In this case, prior values are changed for dogwhelks and periwinkles, as these are directly manipulated, but not for other nodes (see S3 File for an example based on the rocky shore model).

Changes in law and policy can also be included in model scenarios. In the MPA management scenario a potting ban was implemented by reducing the lobster fishery (setting the prior to -4), and a total fishing ban to setting both lobster and finfish fisheries nodes to -4. Where policies are thought to be weak or ineffective, values other than +/- 4 can be used to indicate this inherent weakness in the policy.

It is possible to include policy nodes when building models, and these nodes can be connected to relevant model nodes via edges. For example, a lobster fishing ban policy node could be linked to the lobster fishing node with an edge set to -4. This approach can be beneficial in complex policy scenarios or when multiple nodes change due to policy implementation. However, it is generally simpler to represent the effects of policies directly in a basic model by adjusting the priors.

The '*BBNet*' model prevents signal loss (Eqs 3 and 4 in section 2), meaning that priors in the `bbn.predict()` function only change when they become more certain (i.e. closer in value to 4 or -4). This means that when a prior is intended to be important measured outcomes of the model, the process to prevent signal loss may limit the model's ability to determine the overall influence of the system on that particular prior node. In these cases, developing a policy node, as described earlier, allows for the examination of all nodes (other than policy nodes). For example, implementing a lobster fishing ban in the MPA system could lead to an increase in lobster population, which might result in more illegal lobster fishing. If the lobster fishing ban is implemented by simply changing the lobster fishing node prior to -4, the model would

be unable to predict this increase in illegal fishing, as such activity would move this node towards zero (indicating less certainty). However, having a policy node for the lobster fishing ban set to -4 as a prior, connected to the lobster fishing node with an edge of -4, would prevent the policy node from changing, while allowing the lobster fishing node to increase to reflect the rise in illegal fishing.

## 4.5 Involving others in building the model

The relative intuitiveness of the network model approach does lend itself to a collaborative model and scenario building process. Indeed, while a framework for scoring interaction strengths has been given; for models which are going to be used beyond the scope of hypothesis development, it is useful to have multiple people involved in designing and parametrising models. BBNs can be built from expert opinion. Processes such as the Delphi method can be used to obtain agreement on nodes and edges [25], and potentially even interaction strength. Disagreements can be resolved by assigning disputed edge strengths based on data or literature (as per Table 2) or assessing the importance of the interaction under question using the sensitivity analysis functions. Stakeholder groups can also inform nodes, edges and edge strengths in the BBNs [18].

Stakeholder interaction and consultation can also be useful for refining the models and ensuring maximum trust in the model outputs [26]. In particular, stakeholders may have views different from scientific experts or scientific literature on some topics (e.g. the effects of fishing [27]). Such disagreements may involve building two or more models to compare the results of these disagreements. Stakeholders can also design scenarios for exploration, based on how policy, management, environmental conditions etc. may affect the system being considered. Given the relative ease of creating scenarios, it may be possible to produce and analyse these in real time in meetings with stakeholder groups.

Another application of BBNs is in the aiding of transfer of knowledge between academics and practitioners (e.g. government policy makers). These models, even if quickly produced, can facilitate dialogue between academic knowledge and potential implications and consequences of policy formation [20]. They can also be tailored to specific requirements and outcomes. Using BBNs as a mechanism for information transfer between academic and practitioner sectors may facilitate some of the difficulties currently faced in these knowledge exchange activities [28].

## 5. Conclusions

We have presented an approach to predictive ecological and environmental modelling (which can link to social science outcomes e.g. [15, 18]) which can be rapid to develop, easy to use (albeit with some degree of training or troubleshooting support) and intuitive to understand key concepts and outcomes, particularly for non-specialists including policy makers and NGOs. The methodological overview presented here and the R package functions for the 'BBNet' package provide a framework for the use of these models and a user-friendly interface for creating and analysing the models.

BBNs models will not fulfil every requirement of current modelling processes, and do not produce fully quantitative data (e.g. estimates of fish biomass in tonnes, or value of ecosystem services in US$). They do, however, allow different scenarios to be explored and evaluated relative to each other [21], predict the direction of change in various parts of a system [16], and handle complex systems with environmental, ecological, and social aspects [20].

Additionally, the 'BBNet' package can account for feedback loops within the system over varying timescales [8]. It can be used to develop hypotheses which can be tested empirically

[8], produce results which inform policy [20], capture community and management group understanding [18], and address concerns and facilitate dialogue with practitioners [18]. However, it can also produce meaningful research outputs in their own right and gain understanding of complex system dynamics.

While we have focussed on the use of the models in environmental problems, their application does not need to be restricted to this, and use in financial systems, molecular biology, political sciences, and many other disciplines are likely possible (as an example, the '*BBNet*' package has been used to model the high level financial and political landscape of the UK and predict how different outcomes of the 2024 general election would change this landscape [29]).

## Supporting information

**S1 File. Input files for edges and priors for each of the two example models.** See section 3.1 for further details.
(ZIP)

**S2 File. Tutorial video of input file requirements for the '*BBNet*' package.**
(MP4)

**S3 File. Tutorial video of running '*BBNet*' functions using provided data files.**
(MP4)

**S4 File. R Markdown script with necessary functions and example R code to import input files and run '*BBNet*' functions.**
(RMD)

**S5 File. Tutorial video on incorporating more complex probability matrices in '*BBNet*' through model structure.**
(MP4)

## Author Contributions

**Conceptualization:** Abigail R. Croker, Richard Stafford.

**Data curation:** Richard Stafford.

**Investigation:** Abigail R. Croker, Richard Stafford.

**Methodology:** Victoria Dominguez Almela, Abigail R. Croker, Richard Stafford.

**Resources:** Victoria Dominguez Almela, Richard Stafford.

**Software:** Victoria Dominguez Almela, Richard Stafford.

**Validation:** Victoria Dominguez Almela.

**Visualization:** Victoria Dominguez Almela.

**Writing – original draft:** Victoria Dominguez Almela, Abigail R. Croker, Richard Stafford.

**Writing – review & editing:** Victoria Dominguez Almela, Abigail R. Croker, Richard Stafford.

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
