## [Decision Letter · Decision Letter 0]

22 Jul 2024

PONE-D-24-20910Creating simple predictive models in ecology, conservation and environmental policy based on Bayesian belief networksPLOS ONE

Dear Dr. Stafford,

Thank you for submitting your manuscript to PLOS ONE. After careful consideration, we feel that it has merit but does not fully meet PLOS ONE’s publication criteria as it currently stands. Therefore, we invite you to submit a revised version of the manuscript that addresses the points raised during the review process.

We look forward to receiving your revised manuscript.

Kind regards,

Abroon Qazi, Ph.D.

Academic Editor

PLOS ONE

Journal Requirements:

Reviewers' comments:

Reviewer's Responses to Questions

**Comments to the Author**

1. Is the manuscript technically sound, and do the data support the conclusions?

Reviewer #1: Partly

2. Has the statistical analysis been performed appropriately and rigorously? 

Reviewer #1: Yes

3. Have the authors made all data underlying the findings in their manuscript fully available?

Reviewer #1: Yes

4. Is the manuscript presented in an intelligible fashion and written in standard English?

Reviewer #1: No

5. Review Comments to the Author

Reviewer #1: Overall: The authors efforts in conducting the research and writing the manuscript are appreciated. However, the paper needs significant reorganization to ensure consistency with its aims and content, as well as a stronger methodology section. I believe developing a new version of the paper would be a better option.

The paper lacks a strong structure. Specifically, it is missing an abstract that summarises the objectives, methods, results, and conclusions. While the paper includes an introduction, it does not provide adequate background information and context, state the research problem or question, explain the significance and objectives of the study. Furthermore, it is highly recommended to include a literature review section that examines relevant previous research, identifies gaps in the existing literature, and demonstrates how the current research builds upon or differs from past studies. Additionally, the paper lacks both methodology and results sections, making it difficult for readers to fully understand the research.

Abstract: The abstract should clearly state the primary objective and contribution of the work, highlighting the significance of the new R package (BBNet) earlier on. It should better articulate how the proposed approach simplifies traditional models and clarify what is meant by the models not being fully quantitative. Additionally, the abstract should briefly describe the parameterisation process, mention specific examples of the model's application across different fields, and suggest potential future research directions or improvements.

Introduction: The introduction is packed with numerous points that make it hard to follow. It lacks a clear, concise narrative, making it overwhelming for readers to understand the amin focus. Also, it jumps between various concepts without clear transitions. However, consider breaking the introduction into more distinct paragraphs, each focusing on a specific point. For example, you could have separate paragraphs for:

• The current state of quantitative skills in the environmental workforce.

• The challenges of using sophisticated models and AI in ecology.

• The introduction and benefits of Bayesian belief networks (BBNs).

• The limitations of traditional BBNs.

• Recent advancements in BBNs and their applications.

The introduction includes technical terms that may not be accessible to all readers. Terms like "probability matrices," "reciprocal feedback," and "signal loss" are not adequately explained.

While the audience is likely to be familiar with ecological and modelling terms, consider defining or briefly explaining more complex concepts, such as Bayesian belief networks and reciprocal feedback, to ensure clarity for all readers.

Engage the reader by highlighting the practical implications of your work early on. For example, mention how the BBNet package could simplify the work of environmental researchers and practitioners, making complex modelling accessible to a broader audience.

Theoretical basis: This section provides a comprehensive overview of Bayesian belief network (BBN) models and introduces the theoretical foundation of the BBNet package. To enhance clarity and structure, it is helpful to begin with a brief introductory sentence summarising the purpose of BBN models.

Conclusion: The conclusion outlines the benefits and drawbacks of using BBN models in predictive ecological and environmental modeling. However, it would be enhanced by addressing the following points:

• The assertion that the approach is "rapid and easy to use and understand" for non-specialists, including stakeholders, may be overstated. Complex models, even those designed to be user-friendly, can pose significant challenges for those without a strong background in the relevant fields.

• The discussion of potential future research directions or applications should be more detailed.

• Ensure there is a logical flow from the summary of findings to implications and future directions.

• Consider using subheadings to organise different points for better readability.

6. PLOS authors have the option to publish the peer review history of their article (what does this mean?). If published, this will include your full peer review and any attached files.

Reviewer #1: No

---

## [Author Response · Author response to Decision Letter 0]

2 Sep 2024

We have left the reviewer’s comments in plain black font, our replies are in italics, and any quotes from the revised text of the paper are presented in red in the uploaded file. This may not show up in the repeated version submitted on the online form

Reviewer #1: Overall: The authors efforts in conducting the research and writing the manuscript are appreciated. However, the paper needs significant reorganization to ensure consistency with its aims and content, as well as a stronger methodology section. I believe developing a new version of the paper would be a better option.

The paper lacks a strong structure. Specifically, it is missing an abstract that summarises the objectives, methods, results, and conclusions. While the paper includes an introduction, it does not provide adequate background information and context, state the research problem or question, explain the significance and objectives of the study. Furthermore, it is highly recommended to include a literature review section that examines relevant previous research, identifies gaps in the existing literature, and demonstrates how the current research builds upon or differs from past studies. Additionally, the paper lacks both methodology and results sections, making it difficult for readers to fully understand the research.

We thank the reviewer for their thorough reading of the paper. The paper presents work based on a new R analysis package, and explains the underlying theory of the package and examples for how it is used. PLoS ONE was chosen to submit to, as such studies are included in the journal’s scope. However, there are no formal results of this study, so it isn’t possible to structure in a traditional manner. We have made a number of changes to the paper based on the reviewer’s suggestions, which are detailed in later sections. 

Abstract: The abstract should clearly state the primary objective and contribution of the work, highlighting the significance of the new R package (BBNet) earlier on. It should better articulate how the proposed approach simplifies traditional models and clarify what is meant by the models not being fully quantitative. Additionally, the abstract should briefly describe the parameterisation process, mention specific examples of the model's application across different fields, and suggest potential future research directions or improvements.

We thank the reviewer for these suggestions. The Abstract now reads:

Predictive models are often complex to produce and interpret, yet can offer valuable insights for management, conservation and policy-making. Here we introduce a new modelling tool (the R package ‘BBNet’), which is simple to use, and requires little mathematical or computer programming background. By using straightforward concepts to describe interactions between model components, predictive models can be effectively constructed using basic spreadsheet tools and loaded into the R package. These models can be analysed, visualised, and sensitivity tested to assess how information flows through the system’s components and provide predictions for future outcomes of the systems. This paper provides a theoretical background to the models, which are modified Bayesian belief networks (BBNs), and an overview of how the package can be used. The models are not fully quantitative, but outcomes between different modelled scenarios can be considered ordinally (i.e. ranked from ‘best’ to ‘worse’). Parameterisation of models can also be through data, literature, expert opinion, questionnaires and/or surveys of opinion, which are expressed as a simple ‘weak’ to ‘very strong’ or 1-4 integer value for interactions between model components. While we have focussed on the use of the models in environmental and ecological problems (including with links to management and social outcomes), their application does not need to be restricted to these disciplines, and use in financial systems, molecular biology, political sciences and many other disciplines are possible. 

Introduction: The introduction is packed with numerous points that make it hard to follow. It lacks a clear, concise narrative, making it overwhelming for readers to understand the amin focus. Also, it jumps between various concepts without clear transitions. However, consider breaking the introduction into more distinct paragraphs, each focusing on a specific point. For example, you could have separate paragraphs for:

• The current state of quantitative skills in the environmental workforce.

• The challenges of using sophisticated models and AI in ecology.

• The introduction and benefits of Bayesian belief networks (BBNs).

• The limitations of traditional BBNs.

• Recent advancements in BBNs and their applications.

We have modified the introduction to the following paragraphs

- The mismatch between the lack of quantitative experience in most environmental scientists and policy makers and the complexity of building and interpreting models (i.e. a problem statement, which our work will address)

- The needs of policy makers and conservationists from predictive models

- The introduction and benefits of Bayesian belief networks (BBNs) 

- The limitations of traditional BBNs.

- Recent advancements in BBNs and their applications.

- Aim of the current study

The introduction includes technical terms that may not be accessible to all readers. Terms like "probability matrices," "reciprocal feedback," and "signal loss" are not adequately explained. While the audience is likely to be familiar with ecological and modelling terms, consider defining or briefly explaining more complex concepts, such as Bayesian belief networks and reciprocal feedback, to ensure clarity for all readers.

We have revised these sections. The term ‘probability matrices’ has been removed and there are either definitions or ecological examples for the other terms used. 

Engage the reader by highlighting the practical implications of your work early on. For example, mention how the BBNet package could simplify the work of environmental researchers and practitioners, making complex modelling accessible to a broader audience.

We have added sections which make this point throughout the introduction, in addition to this sentence, just before the study aim.

As such, the ‘BBNet’ package puts tools for constructing and interpreting ecological models in the hands of a far broader number of environmental scientists and professionals than has previously been the case. 

Theoretical basis: This section provides a comprehensive overview of Bayesian belief network (BBN) models and introduces the theoretical foundation of the BBNet package. To enhance clarity and structure, it is helpful to begin with a brief introductory sentence summarising the purpose of BBN models.

We have modified the introduction to this section to read: Bayesian belief networks (BBNs) are a modelling approach where interactions between different components of complex systems can be examined and predictions can be made for components of interest in these systems, as such, they can be used to make environmental or ecological predictions. This is followed by an ecological example.

Conclusion: The conclusion outlines the benefits and drawbacks of using BBN models in predictive ecological and environmental modeling. However, it would be enhanced by addressing the following points:

• The assertion that the approach is "rapid and easy to use and understand" for non-specialists, including stakeholders, may be overstated. Complex models, even those designed to be user-friendly, can pose significant challenges for those without a strong background in the relevant fields.

We have modified this sentence to read: We have presented an approach to predictive ecological and environmental modelling (which can link to social science outcomes) which can be rapid to develop, easy to use (albeit with some degree of training or troubleshooting support) and intuitive to understand key concepts and outcomes, particularly for non-specialists including stakeholders. We have been using earlier forms of these models with undergraduate students for several years, including the published research demonstrating understanding of key processes by first year undergraduates in reference 22, so we hope this more cautious form of wording is now appropriate. 

• The discussion of potential future research directions or applications should be more detailed.

We have tried to keep the conclusion short, as indicated in the instructions for authors’ section. However, we do point to an additional study which has used the BBNet package to address financial and political processes. While we envisage most readers of this manuscript will be environmentally focussed, and multiple examples of where this work has been used in this context are provided in the manuscript, we hope this additional work will illustrate some of the other uses and applications in other disciplines. 

• Ensure there is a logical flow from the summary of findings to implications and future directions.

• Consider using subheadings to organise different points for better readability.

Response to both points above. As this is a relatively short section, we have not added subheadings, but have broken this up into paragraphs and slightly revised the structure, which now broadly follows:

- We have created a framework for simple and intuitive models (but see point above)

- Limitations of the models

- Advances of the ‘BBNet’ package

- Future use of the models

---

## [Decision Letter · Decision Letter 1]

29 Oct 2024

PONE-D-24-20910R1Creating simple predictive models in ecology, conservation and environmental policy based on Bayesian belief networksPLOS ONE

Dear Dr. Stafford,

Thank you for submitting your manuscript to PLOS ONE. After careful consideration, we feel that it has merit but does not fully meet PLOS ONE’s publication criteria as it currently stands. Therefore, we invite you to submit a revised version of the manuscript that addresses the points raised during the review process.

We look forward to receiving your revised manuscript.

Kind regards,

Abroon Qazi, Ph.D.

Academic Editor

PLOS ONE

Reviewers' comments:

Reviewer's Responses to Questions

**Comments to the Author**

1. If the authors have adequately addressed your comments raised in a previous round of review and you feel that this manuscript is now acceptable for publication, you may indicate that here to bypass the “Comments to the Author” section, enter your conflict of interest statement in the “Confidential to Editor” section, and submit your "Accept" recommendation.

Reviewer #2: (No Response)

2. Is the manuscript technically sound, and do the data support the conclusions?

Reviewer #2: No

3. Has the statistical analysis been performed appropriately and rigorously? 

Reviewer #2: N/A

4. Have the authors made all data underlying the findings in their manuscript fully available?

Reviewer #2: No

5. Is the manuscript presented in an intelligible fashion and written in standard English?

Reviewer #2: Yes

6. Review Comments to the Author

Reviewer #2: General comment

As a kind of probabilistic graphical model BBNs became very popular to scientists and practitioners mainly due to the powerful probability theory involved, which makes them able to deal with a wide range of problems including environmental modelling. In this context the authors present a new modelling tool available as an R package specifically developed for environmental and ecological problems though claiming its applicability to other contexts.

Given the recognized technical complexity of BBNs outside the community of statistical modelers a theoretical introductory section would be beneficial for a valuable comprehension while for the theoretical basis of the presented model the authors redirect the reader to another scientific paper published in 2015 by one of the authors [8].

According to the aim and scope of the Journal new software presentation are welcome provided that a proper validation is included demonstrating that the new tool achieves its intended purpose. This requirement may be met by including a proof-of-principle experiment or analysis when possible while the authors focused on presenting two datasets suitable for running the model and spent instead a few section intended as a user-manual likely more suitable for an appendix.

Despite the capabilities of the proposed R code including various functionality which are relevant for complex problem modelling (e.g. sensitivity analysis) the manuscript lack of any test case application thus making the overall presentation hard to be valued by a broader audience. Moreover the xls code presented in [8] seems to have similar predictive features so a comparison to highlight the improvements and/or proven advantage of the new R package over existing alternatives would be interesting too.

Specific comments:

In section 2, Introduction, the BBNet is suddenly introduced here for the first time (except in the abstract) as if it was introduced somewhere before. It would be more readable to introduce it before talking about its features.

In section 2, Theoretical basis, it seems not usual to refer totally to a previous paper if this is relevant for the methodology adopted in this paper. This point was risen also in the general comment. This section has to be thoroughly reviewed as the equations’ part follows the parameterization details making the reading hard and confusing. Moreover concerning the mathematical formalism, the use of the i index for the increasing population may create confusion with the summation symbolism.

Section 3 reports details on the software availability, download repository and installation which are commonly reported after the conclusion section.

Bub-sections 3.1 and 3.2 are more written as a user manual than a science paper. They could be more fitted for an appendix than the body of the paper.

Section 4 is dedicated to the model implementation with its functionalities, features and parameterization. Once again, the paper is hard to follow. Without graphical feature and tables, possibly related to some real case application and validation, the presentation of the BBN model capability may appear minimized.

This is also evident in the Conclusion session where an intriguing sentence on the BBNet capability is hardly supported by the arguments developed in the manuscript. It is hard to justify the reasons for judging the value of this package without any case study with real data.

Moreover, at the end of this section the authors are claiming the use of this model package in high level financial and political landscape of the UK referring to a paper that is not included in the reference list.

7. PLOS authors have the option to publish the peer review history of their article (what does this mean?). If published, this will include your full peer review and any attached files.

Reviewer #2: No

---

## [Author Response · Author response to Decision Letter 1]

20 Nov 2024

Response To Reviewer - please also see attached file which contains correct formatting of these comments

We thank the reviewer for their detailed comments on the manuscript. We are slightly confused as to why this paper was sent for a second round of review to a new reviewer, following us previously addressing another reviewer’s comments and providing a revision. We fully acknowledge the reviewer’s comments regarding the paper lacking a clear set of results, and reading more like an instruction manual. This was the purpose of the manuscript. As noted, previous studies (by some of the authors of this current paper) have used this modelling approach, and it has evolved and added new features over the last 9 years. To make to model as accessible as possible to the academic community, we have produced an R package, with similar predictive features to those used in previous studies, but with additional elements for network visualisation and understanding. The R package, along with this manuscript, and several training courses, should allow a wide uptake of the package for simple modelling tasks. We would argue that the basis of the predictive models has already been validated in previous literature, however the current paper presents an open access, user-friendly way to access this software, a protocol to follow for future studies, and the inclusion of the mathematical basis of the model also allows this manuscript to be a single citable source for the software. It is clear that there may be disagreement in the scope of the paper between the authors and reviewer, and we believe this can only be addressed by an editorial decision on the manuscript. However, we would also like to point out that the scope of the paper has not changed from the initial submission and editorial screening and that similar studies have been published within the journal within the last year – some examples below:

https://journals.plos.org/plosone/article?id=10.1371/journal.pone.0297930

https://journals.plos.org/plosone/article?id=10.1371/journal.pone.0299993

https://journals.plos.org/plosone/article?id=10.1371/journal.pone.0309210

We would therefore question why a manuscript this far along in the review process would therefore be considered unsuitable for the journal. 

While we acknowledge the difference in scope of the article we have produced and what the reviewer has suggested, which does cover a number of the comments below, we also address each comment in turn (our response in italics). 

Reviewer #2: General comment

As a kind of probabilistic graphical model BBNs became very popular to scientists and practitioners mainly due to the powerful probability theory involved, which makes them able to deal with a wide range of problems including environmental modelling. In this context the authors present a new modelling tool available as an R package specifically developed for environmental and ecological problems though claiming its applicability to other contexts.

Given the recognized technical complexity of BBNs outside the community of statistical modelers a theoretical introductory section would be beneficial for a valuable comprehension while for the theoretical basis of the presented model the authors redirect the reader to another scientific paper published in 2015 by one of the authors [8].

We appreciate the line in question is poorly written and undermines the current study. As indicated in our overall aims, this paper does present the theoretical framework of the model, so it can be used as single point of reference for future studies. We have rephrased this to read: The theoretical basis for the model is based on that in Stafford et al. (2015) [8] and is described below. A number of updates and additional useful tools are provided in the ‘BBNet’ package, described in the functions below (section 3), which provide additional functionality to understand and visualise the models and to examine information flow through the models. 

According to the aim and scope of the Journal new software presentation are welcome provided that a proper validation is included demonstrating that the new tool achieves its intended purpose. This requirement may be met by including a proof-of-principle experiment or analysis when possible while the authors focused on presenting two datasets suitable for running the model and spent instead a few section intended as a user-manual likely more suitable for an appendix.

We address this point above. However, the scope of the journal in terms of software articles states: “Submissions presenting methods, software, databases, or tools must demonstrate that the new tool achieves its intended purpose. If similar options already exist, the submitted manuscript must demonstrate that the new tool is an improvement over existing options in some way. This requirement may be met by including a proof-of-principle experiment or analysis; if this is not possible, a discussion of the possible applications and some preliminary analysis may be sufficient”. This is different the criteria stated by the reviewer, it is the improvement from a similar technique that may require a proof of principle experiment or analysis. As clearly stated, the aim of our paper is to: present (1) the underlying theory of the modified Bayesian belief networks, (2) introduce the ‘BBNet’ package as a user-friendly interface for ecological and environmental researchers and practitioners with limited modelling experience to produce useful and meaningful models, and (3) suggest a workflow for the formulation of these models, including parameterisation of the model and dealing with uncertainty. We believe the manuscript fulfils these goals. 

Despite the capabilities of the proposed R code including various functionality which are relevant for complex problem modelling (e.g. sensitivity analysis) the manuscript lack of any test case application thus making the overall presentation hard to be valued by a broader audience. Moreover the xls code presented in [8] seems to have similar predictive features so a comparison to highlight the improvements and/or proven advantage of the new R package over existing alternatives would be interesting too.

The reviewer is right, the excel code in previous work has similar predictive abilities, as it is based on the same set of equations. Excel and R code has been used in many studies which have used this framework [8], [15],[16],[18], [20],[21],[29] in the revised reference list, which is not an exhaustive list. These provide context for the research in a wide range of ecological and environmental applications for those looking for test case examples. The purpose of this paper is to detail the new functionality in the R package, which was not present in these papers. 

Specific comments:

In section 2, Introduction, the BBNet is suddenly introduced here for the first time (except in the abstract) as if it was introduced somewhere before. It would be more readable to introduce it before talking about its features.

The first line of the second paragraph of section 2 does mention the BBNet package when it could be more general, and we have rephased this. However, the manuscript does clearly introduce the BBNet package in the introduction, prior to specifics of the features being introduced in section 2.

In section 2, Theoretical basis, it seems not usual to refer totally to a previous paper if this is relevant for the methodology adopted in this paper. This point was risen also in the general comment. 

See above – this was poorly phrased and has now been addressed. 

This section has to be thoroughly reviewed as the equations’ part follows the parameterization details making the reading hard and confusing.

The equations do follow the details of what nodes and edges are, and the values they can take. This is essential information to give before the equations of the model. We do acknowledge that Table 1 does include some parameterisation information, and does come before the equations. However, table 1 also serves to illustrate the values the model can take. We would also argue that to achieve the main goals of the paper, most readers are unlikely to spend long looking at the mathematical equations, where as the details of the input parameters are essential for building and using the model, and should be placed first. 

 Moreover concerning the mathematical formalism, the use of the i index for the increasing population may create confusion with the summation symbolism.

We appreciate this point. However, while ‘i’ is common in summation, other values such as x and n are also frequently used. Using ‘i’ for increasing and ‘d’ for decreasing, and n for summation in our equations does seem logical, and we do clearly state: “With subscripts i and d indicate increasing or decreasing respectively for the nodes.”

Section 3 reports details on the software availability, download repository and installation which are commonly reported after the conclusion section.

Bub-sections 3.1 and 3.2 are more written as a user manual than a science paper. They could be more fitted for an appendix than the body of the paper.

As indicated above, we do appreciate this comment and acknowledge it, but this ‘user manual’ approach is our intention in the paper.

Section 4 is dedicated to the model implementation with its functionalities, features and parameterization. Once again, the paper is hard to follow. Without graphical feature and tables, possibly related to some real case application and validation, the presentation of the BBN model capability may appear minimized.

We agree that this could be made clearer. However, given the purpose of the paper, there is a tradeoff between being general (and therefore more useful as a longer-term resource) and being specific, and thus harder to generalise to readers’ own modelling needs. Figure 1 does show the Rocky Shore network model, and we now refer back to this in this section to clarify some of the points made. We also refer to supplementary material 3 again, which provides a run through of the model and discusses outcomes including graphical and numerical outputs. We hope this provides the necessary context for those who would require this. 

This is also evident in the Conclusion session where an intriguing sentence on the BBNet capability is hardly supported by the arguments developed in the manuscript. It is hard to justify the reasons for judging the value of this package without any case study with real data.

We appreciate that the manuscript itself doesn’t provide evidence for all these claims. As such, we have cited literature where this modelling approach has been used in these contexts. 

Moreover, at the end of this section the authors are claiming the use of this model package in high level financial and political landscape of the UK referring to a paper that is not included in the reference list.

This has been added. We appreciate this is a pre-print, it is currently under consideration for publication, also in PLoS ONE. Nevertheless it does illustrate the points of integrating multiple complex systems.

---

## [Editor Report · Decision Letter 2]

25 Nov 2024

Creating simple predictive models in ecology, conservation and environmental policy based on Bayesian belief networks

PONE-D-24-20910R2

Dear Dr. Stafford,

We’re pleased to inform you that your manuscript has been judged scientifically suitable for publication and will be formally accepted for publication once it meets all outstanding technical requirements.

Kind regards,

Abroon Qazi, Ph.D.

Academic Editor

PLOS ONE

Additional Editor Comments (optional):

Thank you for your patience. A new reviewer was assigned for the latest review round because the original reviewer from the first round was unable to complete their evaluation.
---

## [Editor Report · Acceptance letter]

29 Nov 2024

PONE-D-24-20910R2 

PLOS ONE

Dear Dr. Stafford, 

I'm pleased to inform you that your manuscript has been deemed suitable for publication in PLOS ONE. Congratulations! Your manuscript is now being handed over to our production team.

Kind regards, 

on behalf of

Dr Abroon Qazi 

Academic Editor

PLOS ONE